# Enhanced FRER Mechanism in Time-Sensitive Networking for Reliable Edge Computing

**DOI:** 10.3390/s24061738

**Published:** 2024-03-07

**Authors:** Shaoliu Hu, Yueping Cai, Shengkai Wang, Xiao Han

**Affiliations:** School of Microelectronics and Communication Engineering, Chongqing University, Chongqing 400030, China; caiyueping@cqu.edu.cn (Y.C.); wangshengkai@stu.cqu.edu.cn (S.W.); han8060@163.com (X.H.)

**Keywords:** time-sensitive networking, edge computing, frame replication and elimination for the reliability, path reliability, edge-disjoint path pair

## Abstract

Time-Sensitive Networking (TSN) and edge computing are promising networking technologies for the future of the Industrial Internet. TSN provides a reliable and deterministic low-latency communication service for edge computing. The Frame Replication and Elimination for Reliability (FRER) mechanism is important for improving the network reliability of TSN. It achieves high reliability by transmitting identical frames in parallel on two disjoint paths, while eliminating duplicated frames at the destination node. However, there are two problems with the FRER mechanism. One problem is that it does not consider the path reliability, and the other one is that it is difficult to find two completely disjoint path pairs in some cases. To solve the above problems, this paper proposes a method to find edge-disjoint path pairs considering path reliability for FRER in TSN. The method includes two parts: one is building a reliability model for paths, and the other one is computing a working path and a redundant path with the Edge-Disjoint Path Pairs Selection (EDPPS) algorithm. Theoretical and simulation results show that the proposed method effectively improves path reliability while reducing the delay jitter of frames. Compared with the traditional FRER mechanism, the proposed method reduces delay jitter by 15.6% when the network load is 0.9.

## 1. Introduction

The rapid development of the Industrial Internet has facilitated the digital transformation of conventional industries [1]. Time-critical real-time applications [2], such as real-time industrial control, autonomous driving, and vehicle-to-vehicle communication, impose stringent requirements on the Industrial Internet, including low latency, low jitter, and high reliability [3]. However, the storage and forwarding method, as well as the flow transmission mechanism of traditional Ethernet technology, fail to meet the Quality of Service (QoS) demanded by these time-critical real-time applications. For instance, industrial control systems necessitate the real-time transmission of substantial volumes of data, and have exceptionally high demands for data accuracy and reliability. Network failures lead to production line downtime and result in significant economic losses. Although traditional Industrial Ethernet and fieldbus [4,5] solutions offer real-time and high reliability guarantees for data transmission, achieving interoperability among different systems remains a challenge [6].

The massive amount of data generated by the Industrial Internet of Things (IIoT) poses significant challenges to data centers and cloud computing. To address this problem, edge computing [7,8,9] emerged. Edge computing shifts the tasks of data processing and analysis from centralized data centers to the edge of the network, enabling time-critical data to be analysed at the source. It not only improves the efficiency of data processing but also allows for the allocation of network resources closer to users. Furthermore, it can facilitate the fulfillment of users’ requirements for low latency, low jitter, and high reliability.

Using edge computing and TSN as tools for IIoT services is a promising solution. It is also promising for TSN to be integrated with cloud computing and fog computing and leverage their respective strengths. Gomez et al. [10] proposed several strategies that combine TSN and edge computing (EC) to ensure low latency, scalability, and interoperability for critical IIoT services. The paper also evaluated these strategies under different congestion levels. However, the balance between the number of edge nodes and the network topology cost need to be considered for these strategies. Yu et al. [11] and Peng et al. [12] have studied the problems of Orthogonal Frequency Division Multiple Access (OFDMA)-based Multi-Access Edge Computing (MEC) schemes. The former proposed an iterative algorithm to determine the data offloading ratio and optimize resource allocation strategies but it lacks comparison with existing excellent solutions. The latter designed a seamless working solution. The solution utilized TSN controllers and MEC coordinators to achieve global forwarding and edge collaboration. This solution lacks the support of experimental or simulation data. Yang et al. [13] built a TSN-testing solution for a real-time edge computing platform. The platform installed communication devices based on TSN to capture and monitor real-time TSN flows. However, this method dose not consider the situation of the mixed transmission of multiple sets of priority traffic during the testing phase. Wang et al. [14] proposed an IndustEdge solution for Edge–Cloud Collaborative Intelligent (ECCI) platforms. The solution used TSN as the link-layer deterministic transmission network to reduce latency for the ECCI platform. But the actual deployment cost of this solution is relatively high. Additionally, Raagaard [15], Barzegaran [16], and Paul [17] have studied the integration or reconstruction of TSN with fog computing platforms. They mainly focused on solving problems from a scheduling perspective. However, they all ignore the feasibility of time synchronization and reliability solutions.

TSN is a set of standards specified by the IEEE 802.1 [18] working group. The working group is dedicated to providing real-time and reliable capabilities to standard Ethernet, achieving cross-domain flexibility, low latency, and high reliability for Industrial Ethernet technologies [19,20,21]. In order to meet the high-reliability and low-latency service requirements for flow transmissions in industrial networks, TSN enhances Ethernet by synchronization, deterministic flow scheduling, and seamless redundancy [22]. IEEE 802.1AS [23] enhances the Precision Time Protocol (PTP) and proposes the generalized Precision Time Protocol (gPTP) for layer 2 networks to achieve precise time synchronization. IEEE 802.1Qbv [24] solves the problem of frame scheduling by introducing Time-Aware Shapers (TAS) and gate-controlled queues using Time-Division Multiple Access (TDMA) technology. To guarantee the bounded latency of frame transmissions, IEEE 802.1Qch [25] introduces a Cyclic Queuing and Forwarding (CQF) mechanism to achieve the synchronized transmission of frames in odd and even cycles. In addition, IEEE 802.1CB [26] proposes the FRER mechanism, which uses disjoint and redundant paths [27] to transmit copies of multiple frames, improving the reliability of data transmission.

In order to ensure the stable operation and quality of service of edge computing, a series of measures need to be taken to improve its reliability. As shown in Figure 1, the edge computing network supports interconnection between edge computing nodes. The Remote Radio Unit (RRU) is connected to the Distribution Unit (DU) and Centralized Unit (CU) [28] through a mobile fronthaul network [29], while the DU and CU are connected to the core network through a mobile backhaul network [30]. The core network provides various network services to users through the data center network. Edge computing network distributes computing resources to each node of the network, providing users with various new applications and services. However, edge computing involves the transmission of a large number of sensitive data and key services, which requires a high reliability. In order to meet the reliability requirements of edge computing networks, TSN is utilized to support the deterministic transmissions of data in edge computing networks.

High reliability is an important characteristic of TSN [31]. The IEEE 802.1CB specifies a new scheme, called FRER, for enhancing reliability in real-time Ethernet networks. This scheme achieves the high-reliability transmission of time-critical flows by sequentially numbering and replicating each time-sensitive frame in the source and relay systems. It also eliminates duplicated frames [32] in the destination and other relay systems, as illustrated in Figure 1. The FRER mechanism achieves the redundancy of critical frames [33,34,35,36] by replicating frames at the source and transmitting them in parallel through disjoint paths. Duplicated frames can still be received at the destination node, even if the original frame cannot reach the destination node. A mechanism for eliminating duplicated frames is provided when both frames reach the destination. This mechanism in TSN provides a high reliability for frame transmissions by providing redundant paths. Additionally, it also has fast fault detection and recovery capabilities. It provides seamless redundant switching to ensure the continuity and reliability of network services. However, this usually also increases bandwidth consumptions.

Despite utilizing spatial redundancy to achieve seamless redundancy [37,38] in frame transmission within the network, FRER still faces great challenges such as cost, timing, path computation, node selection, and schedulability. The computation of working path and redundant path is a core component of FRER, allowing FRER to utilize spatial redundancy. However, traditional FRER mechanisms employ shortest path algorithms to compute working paths and redundant paths. On one hand, this approach calculates paths with the minimum number of hops, without considering the states of links and nodes. As a result, the reliability of the computed working path and redundant path is significantly reduced. On the other hand, this method aims to find paths that avoid intersections with multiple edges and nodes, posing a significant challenge in complex and dynamic network topology.

The contributions of this paper are as follows: (1) This paper utilizes a path reliability model that takes into account the attributes and states of both links and nodes in the path. The model assigns a reliability value to the edge weights in the network graph in an additive form, resulting in the selection of high-reliability paths. (2) This paper proposes an algorithm for selecting edge-disjoint path pairs. This algorithm enables the identification of such path pairs in any network topology and state, as long as edge-disjoint paths exist in the network.

The remaining parts of this paper are structured as follows. In the second section, the related work on FRER is introduced. The third section describes the FRER-EDPPS mechanism proposed in this paper. The FRER-EDPPS mechanism includes the path reliability model, the calculation of edge-disjoint path pairs, and a detailed illustrative example. The fourth section provides theoretical analysis and simulation evaluation of FRER-EDPPS, with evaluation metrics including path reliability, bandwidth consumption ratio, and delay jitter. Finally, the fifth section concludes the paper.

## 2. Related Works

FRER reduces the network’s tolerance to permanent faults by utilizing spatial redundancy. Spatial redundancy becomes unavailable when permanent faults occur. Alvarez et al. [39] introduced temporal redundancy as a means to tolerate transient faults in the channel, and presented two active frame replication schemes to enhance the network’s tolerance to both permanent and transient faults. However, they did not consider link differences and the forwarding mechanism of replicated frames. Feng et al. [40] extended the static offline method of FRER to achieve online incremental rerouting and rescheduling. They also utilized spatial and temporal redundancy to handle both permanent and temporary faults. However, this method lacks a description of the error control mechanism and its complex assumptions present significant challenges to the worst-case delay analysis.

The FRER redundancy scheme poses great challenges to analyzing the timing sequence of TSN. Thomas et al. [41] pointed out that replicating packets through the network leads to a burst increase in the number of duplicate packets on the path. They suggested that rearranging the packets before they reach the elimination node with the regulator can mitigate the increase in worst-case delay bounds. However, this method overlooks the negative impact of reordering on worst-case delay and delay jitter. Mohammadpour et al. [42] investigated the influence of packet reordering on worst-case delay and delay jitter. They demonstrated that, if the flow remains lossless between the source node and the reordering buffer, the buffer itself will not contribute to an increase in worst-case delay and delay jitter. However, they did not offer a comprehensive solution to mitigate or eliminate the impact of packet reordering on the transmission delay of the flow. The authors developed a calculus to compute the reordering late time offset (RTO) for a flow path. The rationality of this calculus was proven through theoretical analysis. However, this paper mainly solves the problem of path selection in the FRER mechanism and proposes a method for finding redundant path pairs.

The reliability of the network is closely related to scheduling. Feng et al. [43] proposed an efficient proactive fault-tolerant scheduling algorithm in their paper, which ensured the schedulability of TSN and improved the fault tolerance. However, this paper studies the problem of selecting redundant paths instead of flow scheduling. Huang et al. [44] proposed a new strategy for selecting candidate routing sets to improve reliability while considering the schedulability of time-triggered flows. They extended the shortest multiple-path routing method to improve reliability, and used a heuristic algorithm based on cost functions to find solutions with higher schedulability for the network. Zhou et al. [45] noticed that previous research did not consider the transient faults in reliability-constrained scheduling and routing in TSN. Based on this, they proposed a solution which was directly applicable to the standardized frame replication and elimination protocol specified in IEEE 802.1CB, which improves the reliability of scheduling and routing in TSN. The authors studied how to manage and allocate network resources to meet the requirements of different flows. A highly efficient network with good schedulability and reliability was established. The performance metrics were schedulability and synchronization time. However, this paper focuses on the selection of redundant paths in the FRER mechanism to improve the reliability of flow transmission in the network. In this paper, the performance metrics are path reliability, bandwidth consumption ratio, and delay jitter.

## 3. Edge-Disjoint Path Pair Selection for the Frame Replication and Elimination Mechanism

### 3.1. FRER-MPC

A P-Cycle is a network topology protection method in optical networks that guarantees rapid protection by pre-configuring cycle paths. Cai et al. [46] proposed an enhanced FRER-MPC mechanism. On one hand, this mechanism selects redundant paths in the FRER mechanism by cascading multiple P-Cycles, thereby increasing the probability of successfully finding two disjoint paths. On the other hand, the mechanism proposes a link priority model to compute the working path, effectively enhancing the reliability of the working path.

The FRER-MPC mechanism consists of three parts: a working path calculation, a redundant path calculation, and a selection of FRER nodes.

The first part introduces a link priority model. The model considers link reliability, link load, and link centrality. Link reliability is defined as the probability of a link being fault-free. Link load is defined as the amount of bandwidth used on the link. Link centrality is defined as the number of shortest paths that include the link. The link priority is calculated by the following equation.
(1)$$P_{ij}^{L}=R_{ij}\cdot \left(1-L_{ij}\right)\cdot C_{ij}=\left(1-p_{ij}\right)\cdot \left(1-\frac{B_{ij}^{U}}{B_{ij}^{T}}\right)\cdot \frac{n_{ij}}{\left|S_{ij}\right|}$$
where $P_{ij}^{L}$ is the link priority information, *i* and *j* represent two nodes of the link. $R_{ij}$ is the link reliability. $p_{ij}$ is the probability of link failure. $L_{ij}$ is the link load. $B_{ij}^{U}$ and $B_{ij}^{T}$ are the bandwidth of the link utilized and the total bandwidth of the link, respectively. $C_{ij}$ is the link centrality. $S_{ij}$ is the set of the shortest paths from *i* to *j*. $n_{ij}$ is the number of the shortest paths that include the link between node *i* and node *j*.

The second part is building multiple cascaded P-Cycles and the elimination of the common edges of the P-Cycles. The calculation steps are as follows: find each link that constitutes the working path and find the P-Cycle that contains this link. If the P-Cycle is unavailable, both the working path and the redundant path utilize this link. Next, merge two adjacent P-Cycles to form a new, larger P-Cycle. Third, continue the merging process until there are no shared links between the remaining P-Cycles. In this way, a redundant path that intersects minimally with the working path is found.

The third part selects appropriate frame replication and elimination nodes. In the FRER mechanism, replication nodes are TSN nodes with the ability to replicate data packets. They replicate and transmit duplicated packets to achieve the redundant transmission of data flows, with the source node being selected as the replication node. Elimination nodes are TSN nodes with the ability to discard data packets. Relay nodes or destination nodes eliminate duplicated frames by identifying the sequence numbers carried by the data frames. The position of the relay node is preferably closer to the destination node. In addition to the source node and the destination node, the common nodes between the working path and the redundant path also have the ability to replicate and eliminate frames. Therefore, the FRER-MPC mechanism has a higher frame processing overhead than the FRER mechanism.

### 3.2. Problems of the FRER-MPC

The network topology is represented by an undirected graph G={V,E}, where *V* represents the set of nodes in the network, *E* represents the set of links, *P* represents the paths, and the source node and destination node are represented by s and d, respectively. In Time-Sensitive Networking, traffic is divided into Time-Triggered (TT) flows, Audio–Video Bridging (AVB) flows, and Best-Effort (BE) flows. For TT flows, which are high-priority periodical time-sensitive flows, the FRER mechanism improves flow transmission reliability through spatial redundancy. While the flow is being transmitted through the working path $P_{w}$, it is also transmitted in parallel through redundant paths $P_{r}$ that do not intersect with the working path, ensuring that the flow can still reach the destination node on time if any link in working path fails. However, the FRER mechanism and FRER-MPC mechanism have the following problems:

(1) The reliability of transmission along a path is related to the status of links and nodes in the path. The FRER mechanism achieves the minimum number of hops in path calculation, resulting in low reliability and high congestion in the working path. It reduces the reliability and efficiency of flow transmissions. The FRER-MPC mechanism calculates a path with a slightly higher reliability but it only considers the attributes of links and nodes in the working path.

(2) The FRER mechanism calculates the working path using the shortest path algorithm. It calculates the redundant path by removing the working path from the network topology. The FRER-MPC mechanism searches for P-Cycles for all of the links in the working path, and it calculates the redundant path by cascading all P-Cycles while merging common edges. On one hand, the FRER mechanism emphasizes that the redundant path should not intersect with the working path, which may result in the failure to find the redundant path. On the other hand, when there are common edges on the working path between two P-Cycles, both the FRER and FRER-MPC mechanisms cannot find hidden intersecting path pairs, as shown in Figure 2b,c.

The redundant paths can only share some links with the working path, greatly increasing the failure-occurrence rate of the transmission path. The FRER-EDPPS method proposed in this paper can find disjoint paths by splitting the working path.

### 3.3. Proposed FRER-EDPPS Mechanism

#### 3.3.1. Path Reliability Model

In an undirected network topology graph G={V,E}, the bandwidth of the link (i,j) is represented by $B_{ij}$, and the ratio of the consumed bandwidth to the total capacity of the link (i,j) is represented by $\alpha_{i,j}$. $p_{i,j}$ and $q_{i,j}$ represent the probability of link (i,j) failure and the probability of the normal operation of the link (i,j), respectively. The probability of node *i* failure and the probability of the normal operation of node *i* are represented by $p_{i}$ and $q_{i}$, respectively. The cache of node *i* is represented by $C_{i}$, and the cache utilization of node *i* is represented by $\beta_{i}$. $P_{w}$ and $P_{r}$ represent the working path and the redundant path, respectively, and they do not have common edges.

Traditional path calculation methods only calculate the shortest path with the fewest hops, resulting in a low path reliability. The transmission and delivery of time-critical flows are easily affected by link and node failures, especially in factories with a harsh environment. It is necessary to fully consider the reliability of nodes and links when selecting paths to meet the flow transmission requirements of time-critical applications. Here, we create a reliability model for paths that considers link load, link reliability, node load, and node reliability. The link load is represented by the bandwidth utilization of the link, and the link reliability is represented by the probability of the normal operation of the link. The node load is represented by the cache utilization of the node, and the node reliability is represented by the probability of the normal operation of the node.

The reliability $R^{L}$ of traffic transmission in a link is related to the state and attributes of the link. The state of the link is represented by the link load, and the higher the bandwidth utilization $\alpha_{i,j}$ of the link (i,j), the greater the probability of congestion and packet loss in the link. The attributes of the link are represented by the link reliability, which is influenced by factors such as link material, location, and environment. The probability of normal operation of the link (i,j) is represented by $q_{i,j}$. The reliability $R^{L}$ of traffic transmission in a link can be calculated using Equation (Equation 2).
(2)$$R_{i,j}^{L}=\left(1-\alpha_{i,j}\right)\cdot q_{i,j}=\left(1-\frac{B_{i,j}^{U}}{B_{i,j}^{T}}\right)\cdot \left(1-p_{i,j}\right),\left(i,j\right)\in E$$
where $R_{i,j}^{L}$ represents the reliability of the link (i,j), $B_{i,j}^{U}$ and $B_{i,j}^{T}$ represent the used bandwidth and total bandwidth of the link (i,j), respectively.

The reliability $R^{N}$ of traffic transmission in a node is related to the states and attributes of the node. The state of the node is represented by the cache, and the higher the cache utilization $\beta_{i}$ of node *i*, the greater the probability of congestion and packet loss in the node. The attributes of the node are represented by the node reliability, software stability, environment, management, and other conditions. The reliability $R^{N}$ of traffic transmission in a node can be calculated using Equation (Equation 3).
(3)$$R_{i}^{N}=\left(1-\beta_{i}\right)\cdot q_{i}=\left(1-\frac{C_{i}^{U}}{C_{i}^{T}}\right)\cdot \left(1-p_{i}\right),i\in V$$
where $R_{i}^{N}$ represents the reliability of node *i*, $C_{i}^{U}$ and $C_{i}^{T}$ represent the used cache and total cache size of node *i*, respectively.

The reliability $R^{P}$ of traffic transmission in a path is composed of both link and node aspects. Therefore, the reliability of traffic transmission in a path can be calculated using Equation (Equation 4).
(4)$$R^{P}=\prod_{\left(i,j)\in E(P\right)}R_{i,j}^{L}\cdot \prod_{k\in V(P)}R_{k}^{N}$$

To obtain the path with the highest reliability, all of the links and the nodes at both ends of the links in the network are considered as paths. The reliability of the path is used as the metric to assign values to all edges in the network topology graph, and the best-performing working path and redundant path are selected. Since Dijkstra’s algorithm is used in the calculation of the working path and redundant path, and the edges in the path are connected in order, the reliability metrics are multiplied together. Therefore, the reliability metrics need to be transformed from a multiplication form to an addition form, and then the edge weights in the network topology graph can be assigned accordingly. In this way, finding the shortest path in the weighted graph *G* is equivalent to finding the path with the highest reliability. Equation (Equation 5) provides the processing method.
(5)$$W_{i,j}=-\ln\left(R_{i,j}^{P}\right),\left(i,j\right)\in E$$
where $W_{i,j}$ represents the weight of the edge in the graph G={V,E}. Through processing, the reliability attribute of the path is presented on the edge weight in an additive form in the network graph *G*. The smaller the edge weight of the selected path, the greater the reliability of the path. For example, the weighted sum of each edge in the working path $P_{w}$ can be calculated by Equation (Equation 6).
(6)$$W_{P_{w}}=\sum_{\left(i,j\right)\in E\left(P_{w}\right)}W_{i,j}=-\ln\left(\prod_{\left(i,j\right)\in E\left(P_{w}\right)}R_{i,j}^{P}\right)=-\ln\left(R^{P_{w}}\right)$$

The reliability $R^{P_{w}}$ of the working path $P_{w}$ is calculated as follows:(7)$$R^{P_{w}}=e^{-W_{P_{w}}}$$

The above equation indicates that, the smaller the selected path weight sum, the higher the reliability of the path. Therefore, the shortest path obtained using the Dijkstra algorithm is the most reliable path.

#### 3.3.2. Edge-Disjoint Path Pair Selection

There are two key points in selecting the two highest reliability edge-disjoint path pairs: First, using reliability-related parameters as measures to achieve optimal reliability of the current path. Second, enhancing the algorithm’s ability to find edge-disjoint path pairs. FRER fails to achieve the former but can satisfy the latter in most network topologies. FRER-MPC partially enhances both of these key points.

The proposed method, FRER-EDPPS, achieves optimal path reliability through a path reliability model and an enhanced path calculation ability by using network topology constraints. The proposed method assigns edge weights to all edges in the graph G={V,E} based on the edge weight function mentioned in Section 3.3.1. The Dijkstra algorithm is used to calculate the shortest path from the source node *s* to the destination node *d*, denoted as $P_{1}$. $P_{1}$ represents the highest reliability path from *s* to *d* in graph *G*. This path is then used as the working path $P_{w}$.

The process of finding redundant paths that do not intersect with the working path can be divided into the following three steps, where the determination of network topology constraints is crucial.

Firstly, remove all edges of the path $P_{w}$ in the graph G={V,E} to obtain a new network topology $G_{1}=G-E\left(P_{w}\right)$. Secondly, use the Dijkstra algorithm to find the shortest path from the source node *s* to the destination node *d* in the graph $G_{1}$. If the result can be calculated, it will be considered as a redundant path $P_{r}$. Finally, if a path from the source node *s* to the destination node *d* cannot be found in the graph $G_{1}$ (i.e., the graph $G_{1}$ is not connected between *s* and *d*), reverse all edges in the working path $P_{w}$ and add them to the graph $G_{1}$ to obtain a new network topology $G_{2}$.

The next step is to judge the network topology constraint: whether it is a disconnected graph $G_{2}$ between *s* and *d* or not. If it is not, two new non-intersecting paths, $P_{3}$ and $P_{4}$, can be found as the working path and the redundant path, respectively.

$P_{3}$ and $P_{4}$ can be calculated as follows. Let $K_{sd}$ be the path from node *s* to node *d*, and let the intersection between $K_{sd}$ and the reversed path of $P_{w}$ be the path $K_{mn}$. Then, the path $K_{mn}$ is a sub-path of $K_{sd}$ and the reversed path of $P_{w}$. And the complement paths to path $K_{mn}$ on path $K_{sd}$ are $K_{sm}$ and $K_{nd}$. The complement paths to path $K_{mn}$ on the reversed path of $P_{w}$ are $P_{d}m$ and $P_{n}s$. Among them, the paths $K_{sm}$ and $P_{d}m$ can form a new path $P_{3}$ from node *s* to node *d*, and the paths $K_{nd}$ and $P_{n}s$ can form a new path $P_{4}$ from node *s* to node *d*. $P_{3}$ and $P_{4}$ are pairs of non-intersecting paths on graph *G*. $P_{3}$ and $P_{4}$ are used as a new working path $P_{w}$ and a new redundant path $P_{r}$.

The FRER mechanism copies the original frame in the replication node, and then transmits the original frame and the copy frame in parallel on the working path and the redundant path. Finally the mechanism eliminates the redundant frame and ensures the reliable transmission of the time-critical frame between the nodes. In the FRER-EDPPS mechanism proposed in this paper, the working path and the redundant path are not completely disjoint. Therefore, in addition to the source node and the destination node, the common nodes on the two paths also have the ability to replicate and eliminate, which increase the frame processing overhead of the mechanism proposed in this paper.

#### 3.3.3. FRER-EDPPS Algorithm

The algorithm proposed in this paper is shown in Algorithm 1. Firstly, the edge weights of the network topology are assigned based on the processed path reliability model. Then, the working path is calculated using the edge weight function. Finally, if no redundant path is found in the pruned network graph, two new disjoint paths are calculated as the working path and the redundant path based on the restrictive conditions in the network.

This algorithm first calculates the link and node information. *M* and *N* represent the number of links and nodes, respectively, so the number of calculations is O(M+N). Then, the Dijkstra algorithm is used to find the shortest path through a number of calculations of O(M+Nlog(N)). Therefore, the overall time complexity of the algorithm is O(NlogN).
**Algorithm 1** FRER-EDPPS Algorithm**Input:** Network topology *G*, Link failure rate $p_{i,j}$, Node failure rate $p_{k}$, (i,j)∈E,k∈V. Source node *s*, Destination node *d*.**Output:** Working path $P_{w}$, Redundant path *P*, Replication node $N_{r}$, Eliminate node $N_{e}$. 1: FOR (i=1; i<=N; *i*++)2: FOR (j=1; j<=N; *j*++)3: $R_{i,j}^{L}\leftarrow \left(1-p_{i,j}\right)\cdot \left(1-B_{i,j}^{U}/B_{i,j}^{T}\right)$;//Calculate link status and attribute information4: $R_{k}^{N}\leftarrow \left(1-p_{k}\right)\cdot \left(1-C_{k}^{U}/C_{k}^{T}\right)$;//Calculate node status and attribute information5: $R^{P}\leftarrow R_{i,j}^{L}\cdot R_{k}^{N}$;//Calculating path reliability model6: $W_{i,j}\leftarrow -\ln\left(R_{i,j}^{P}\right)$;//Process path reliability model7: END FOR8: END FOR9: $P_{s}=\left\{P_{s1},P_{s2},\ldots ,P_{sk}\right\}\leftarrow Dijkstra\left(s,G,d\right)$;//Calculate the shortest path set10: $P_{w}\leftarrow P_{s}\left\{{\left(W_{i,j}\right)}_{\max}\right\}$;//Select the path with the highest reliability as the working path11: $G_{1}\leftarrow G-E\left(P_{w}\right)$;//Delete all edges of the working path in *G*12: $P_{s}=\left\{P_{s1},P_{s2},\ldots ,P_{sk}\right\}\leftarrow Dijkstra\left(s,G_{1},d\right)$;//Calculate the shortest path set13: IF $\left(P_{s}\neq \emptyset \right)$//There exist a shortest path14: $P_{r}\leftarrow P_{s}\left\{{\left(W_{i,j}\right)}_{\max}\right\}$;//Filter out the path with the highest reliability as a redundant path15: ELSE16: $G_{2}\leftarrow G_{1}+E\left(P_{w(d\to s)}\right)$;//Add the reverse edge set of the work path to the disconnected graph $G_{1}$17: IF $Dijkstra\left(s,G_{2},d\right)\to P_{s}\neq \emptyset$//There are two paths that do not intersect18: $P_{w}^{\mathrm{new}},P_{r}^{new}\leftarrow P_{s}$;19: $P_{w},P_{r}\leftarrow P_{w}^{new},P_{r}^{new}$;//Calculate two disjoint paths and update them with new working and redundant paths20: END IF21: $N_{r},N_{e}\leftarrow Node\left(P_{w}\cap P_{r}\right)$;//Filter common nodes for working and redundant paths22: **RETURN** $P_{w},P_{r},N_{r},N_{e}$;//Output working path, redundant path, copy nodes, eliminate nodes

#### 3.3.4. A Working Example of FRER-EDPPS

As shown in Figure 3, when FRER, FRER-MPC, and the proposed FRER-EDPPS can calculate two disjoint paths, the working path is selected as s-e-f-d. The weight on each link represents the processed additive reliability measure, where smaller numbers indicate more reliable paths. Since FRER prioritizes hop count (as shown in Figure 3b), the redundant path is s-a-b-c-d with a reliability measure of 11. FRER-MPC solves the working path based on cascaded P-Cycles (as shown in Figure 3c), the redundant path is s-a-e-b-f-c-d with a reliability measure of 12. The method proposed in this paper measures based on reliability index (as shown in Figure 3d), the redundant path is s-a-b-f-c-d with a reliability measure of 10.

## 4. Performance Evaluation

This section presents a theoretical and simulation analysis of the FRER-EDPPS mechanism proposed in this article. The simulation analysis evaluates the performance of the shortest path mechanism, FRER mechanism, FRER-MPC mechanism, and FRER-EDPPS mechanism. The performance evaluation metrics include path reliability, bandwidth consumption ratio, and delay jitter. The detection and evaluation of various performance metrics for each mechanism are conducted under different network loads.

### 4.1. Theoretical Evaluation

The traditional FRER mechanism utilizes the shortest path algorithm to compute redundant paths that avoid intersections with edges and nodes. This poses a significant challenge in complex and dynamic network topologies. The FRER-MPC mechanism treats each link in the working path as an individual unit. By identifying the P-Cycle for each link, it becomes possible to discover two paths that connect the endpoints of the link. Ultimately, by cascading the P-Cycles of each link in the working path, it ensures the discovery of an alternative path that connects the source and destination nodes.

The proposed FRER-EDPPS mechanism enhances the reliability of the chosen path by utilizing a path-reliability model. The selection of each link takes into account the bandwidth, failure rate, and attributes of the nodes at both ends. As the reliability of the path is the product of the attributes of each link and node, the path reliability information is processed to make it additive. Consequently, the reliability information of the path serves as the weight for each edge in the network graph *G*, and the shortest path chosen corresponds to the highest reliability.

The FRER-EDPPS mechanism proposed in this paper improves the probability of successfully finding non-intersecting redundant paths by imposing restrictive conditions. If there are shared links on the working path of a P-Cycle, the FRER and FRER-MPC mechanisms are unable to discover path pairs by dividing the current working path. This issue can be resolved by establishing the aforementioned stringent condition. Theorem 1 states that, if the restrictive condition holds, there must exist two new disjoint paths in the network.

**Theorem 1.** 
*In graph G={V,E}, if there is a path $P_{1}$ from source node s to destination node d, the set of edges in $P_{1}$ can be denoted as $E_{1}=\left\{e_{s\to a},\ldots ,e_{b\to d}\right\}$. The cut set of G is $E_{1}=\left\{e_{s\to a},\ldots ,e_{b\to d}\right\}$. Removing the edges in the cut set from G results in a disconnected graph, denoted as $G_{1}=G-E\left(P_{1}\right)$. Reversing all of the edges in $P_{1}$ and adding them to graph $G_{1}$, we obtain a mixed graph $G_{2}=G_{1}+E\left(P_{1}^{re}\right)$, where $P_{1}^{re}$ is the reversed path of $P_{1}$ and its set of edges is $E_{1}^{re}=\left\{e_{d\to b},\ldots ,e_{a\to s}\right\}$. If a path $P^{su}$ exists from the source node to the destination node in graph $G_{2}$, then it is always possible to find two non-intersecting paths $P_{3}$ and $P_{4}$ in graph G.*


**Proof of Theorem 1.** Suppose that a path $P^{su}$ exists from the source node to the destination node in the graph $G_{2}$. The set of edges $P^{su}$ must include some edges from the reverse path $P_{1}^{re}$ (i.e., $E\left(P^{su}\right)\cap E\left(P_{1}^{re}\right)\neq \emptyset$), otherwise $G_{1}$ would be a directed graph, which contradicts the fact that $G_{1}$ is an undirected graph. Therefore, let the intersection of $P^{su}$ and $P_{1}^{re}$ be $P^{in}$, $E\left(P^{in}\right)=\left\{e_{m\to i},\ldots ,e_{j\to n}\right\}$. $P^{in}\neq P_{1}^{re}$, otherwise it contradicts the original condition. So, $P^{in}$ must be non-empty and part of the reverse path $P_{1}^{re}$. Therefore, the set of edges in $P^{su}$ can be represented as $E\left(P^{su}\right)=\left\{e_{s\to x},\ldots ,e_{y\to m},e_{m\to i},\ldots ,e_{j\to n},e_{n\to u},\ldots ,e_{v\to d}\right\}$, removing the set of edges in $P^{in}$ from the sets of edges in $P^{su}$ and $P_{1}^{re}$, resulting in the sets of edges $E_{1}=\left\{e_{s\to x},\ldots ,e_{y\to m}\right\}$, $E_{2}=\left\{e_{n\to u},\ldots ,e_{v\to d}\right\}$, $E_{3}=\left\{e_{d\to b},\ldots ,e_{p\to m}\right\}$, $E_{4}=\left\{e_{n\to q},\ldots ,e_{a\to s}\right\}$. Considering all the edges in the four sets as undirected edges, the combination of $E_{1}$ and $E_{3}$, $E_{2}$, and $E_{4}$ can form two paths $P_{3}$ and $P_{4}$ from the source node *s* to the destination node *d*. $P_{3}$ and $P_{4}$ are paths that do not intersect in terms of edges. The set of edges in $P_{3}$ is $E\left(P_{3}\right)=\left\{e_{s\to x},\ldots ,e_{y\to m},e_{m\to p},\ldots ,e_{b\to d}\right\}$, and the set of edges in $P_{4}$ is $E\left(P_{4}\right)=\left\{e_{s\to a},\ldots ,e_{q\to n},e_{n\to u},\ldots ,e_{v\to d}\right\}$. □

### 4.2. Simulation Evaluation

#### 4.2.1. Performance Metrics

(1) Path Reliability. Evaluating the reliability of a path requires assessing the reliability of its constituent nodes and links. Reliability evaluation metrics are formed using the failure rate, bandwidth consumption rate, node failure rate, and node cache rate of the links. Equation (Equation 8) can be used to calculate the reliability of a path *P*.
(8)RP=∏i,j∈E(P)j∈V(P)(1−pij·μij·γj·pj)
where $R_{P}$ represents the reliability of path *P*. $\mu_{ij}$ represents the ratio of the consumed bandwidth of the link between node *i* and node *j* to the total bandwidth. $\gamma_{j}$ represents the ratio of the cache consumption of node *j* to the total cache size.

The reliability of the working path $P_{w}$ and the redundant path $P_{w}$ can be calculated using Equations (Equation 9) and (Equation 10).
(9)$$R_{P_{w}\cup P_{r}}=R_{P_{w}}+R_{P_{r}}-R_{P_{w}\cap P_{r}}$$
(10)$$R_{P_{w}\cap P_{r}}=R_{P_{w}}\cdot R_{P_{r}}$$

(2) Bandwidth Consumption Ratio. The bandwidth consumption ratio is used to evaluate the utilization of the selected path, which is calculated by dividing the bandwidth consumed by frame transmission by the total bandwidth available. Equation (Equation 11) can be used to calculate the bandwidth consumption ratio.
(11)$$\delta =\frac{B_{u}}{B_{t}}$$
where δ represents the bandwidth consumption ratio, $B_{u}$ represents the bandwidth of frame transmission, and $B_{t}$ consists of the bandwidth of the working path and the bandwidth of the redundant path.

(3) Delay Jitter. End-to-end delay jitter can reflect the quality of the selected path, the quality of links and nodes, and the stability of data transmission. We use $d_{i}$ to represent the delay in the *i*-th data packet in the business flow from the source node to the destination node. The worst-case delay in the business flow can be represented as $\max_{m}\left\{d_{m}\right\}$, which serves as the upper bound of the delay for the business flow. The best-case delay in the business flow can be represented as $\min_{m}\left\{d_{m}\right\}$, which serves as the lower bound of the delay for the business flow. Delay jitter J is defined in RFC3393 as the variation in packet delay, which can be represented by the difference between the upper and lower bounds of the delay, as shown in Equation (Equation 12).
(12)$$J=\max_{m}\left\{d_{m}\right\}-\min_{m}\left\{d_{m}\right\}$$

#### 4.2.2. Network Topology and Simulation Parameters

Figure 4 illustrates the adoption of the European COST-239 [47] as the simulated network topology, while Table 1 presents the traffic models and parameters employed in the simulation.

#### 4.2.3. Simulation Results

As illustrated in Figure 5, the path reliability of the FRER-EDPPS, FRER-MPC, FRER, and SP mechanisms all decrease as the load increases from 0.1 to 1. Among these mechanisms, the SP mechanism has the lowest path reliability due to its lack of frame redundancy transmission. Although the FRER mechanism enhances path reliability through spatial redundancy, it fails to consider the attributes of link and node states. FRER-MPC, which builds upon FRER, improves the reliability of the working path. But the P-Cycle method fails to balance the reliability of redundant paths. Similarly to FRER, FRER-MPC cannot ensure the availability of two disjoint paths for parallel frame transmissions. In contrast, the proposed FRER-EDPPS further enhances path reliability by identifying the pair of disjoint paths with the highest reliability through the path reliability measurement.

As shown in Figure 6, the bandwidth consumption ratios of FRER-EDPPS, FRER, FRER-MPC, and SP mechanisms gradually increase as the load increases from 0.1 to 1. Among the four mechanisms, SP has the lowest bandwidth consumption ratio due to the absence of redundant paths. FRER provides a certain level of reliability for frame transmission through redundant paths, but it also increases the bandwidth consumption compared to the SP mechanism. The proposed FRER-EDPPS in this paper calculates redundant paths based on path reliability measurement, resulting in higher bandwidth consumption compared to FRER, which prioritizes hop count. FRER-MPC improves path reliability by utilizing cascaded P-Cycles. Since the length H of each P-Cycle is set to be greater than or equal to three, FRER-MPC has the highest bandwidth consumption.

The delay jitter of FRER-EDPPS, FRER, and FRER-MPC under different loads is shown in Figure 7. As the load and network congestion increase, the delay jitter of all three mechanisms increases. FRER exhibits inferior path quality compared to FRER-MPC and FRER-EDPPS because it prioritizes working paths and redundant paths based on hop count. When the load exceeds 0.7, the network congestion significantly increases the jitter. Although FRER-MPC performs slightly better in terms of path quality, it also encounters situations where redundant paths cannot be found, making it susceptible to network congestion under high loads. In contrast, the proposed FRER-EDPPS method calculates paths with strong stability. And it can identify redundant paths that do not intersect with edges for the transmission of time-sensitive frames. Additionally, it offers relatively improved path quality.

Figure 8 shows the variation in end-to-end worst-case delay with different proportions of time-critical flows and non-time-critical flows. As the proportion of time-critical flows gradually increases, the worst-case delays in the FRER, FRER-MPC, and FRER-EDPPS mechanisms all decrease. This is because all of the mechanisms have redundant schemes for time-critical flows. The spatial redundancy of time-critical flows improves network congestion problems to reduce the worst-case delay. Furthermore, FRER-MPC and FRER-EDPPS perform better than FRER. This is because both mechanisms consider the path’s properties. The proposed FRER-EDPPS has a better path selection strategy than FRER-MPC.

Figure 9 shows the performance of end-to-end worst-case delay with different frame sizes. As the frame size gradually increases, the worst-case delays in the FRER, FRER-MPC, and FRER-EDPPS mechanisms all increase. This is because the transmission delay in the frame increases as the frame size increases. On the other hand, network congestion may become more severe, and the switching and forwarding nodes require more time to process and forward frames. The FRER-MPC and FRER-EDPPS mechanisms improve congestion conditions by selecting the optimal path, thereby reducing the worst-case delay.

## 5. Conclusions

This paper presents an edge-disjoint path calculation solution to the FRER mechanism in TSN for edge computing. Both the FRER and FRER-MPC mechanism have path pair reliability and existence problems. This paper proposed a calculation method for edge-disjoint path pairs. This method defines a path reliability model to identify the two most reliable edge-disjoint path pairs in the FRER mechanism. Thereby it can achieve the redundant transmission of time-sensitive frames and enhance the reliability of the FRER mechanism. The proposed FRER-EDPPS method consists of the path reliability model and the calculation of edge-disjoint path pairs. The simulation results show that the proposed method can effectively improve path reliability. When the network load is 0.9, compared with traditional FRER and FRER-MPC, the proposed method reduces delay jitter by 15.6% and 11.19%, respectively.

## Figures and Tables

**Figure 1 sensors-24-01738-f001:**
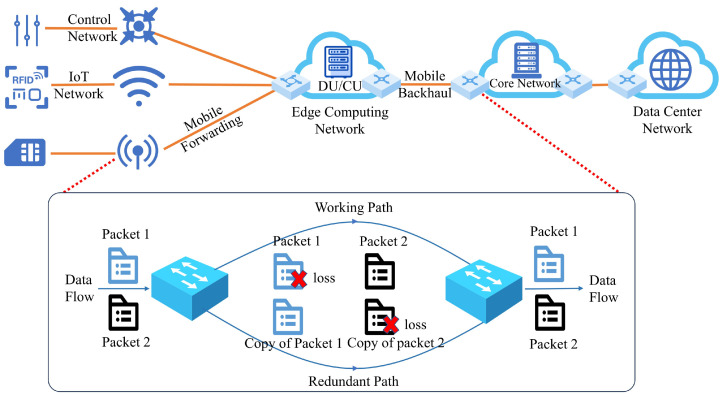
Illustration of FRER mechanism in edge computing network.

**Figure 2 sensors-24-01738-f002:**
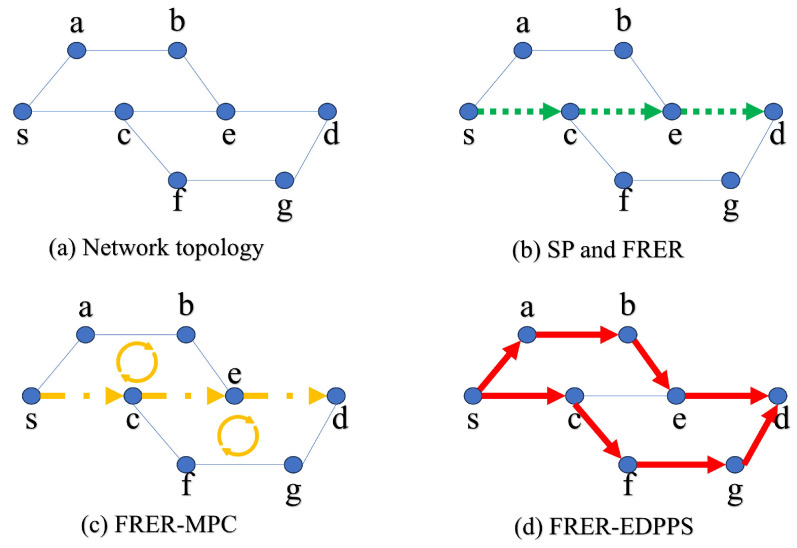
An Illustration Example of FRER-EDPPS.

**Figure 3 sensors-24-01738-f003:**
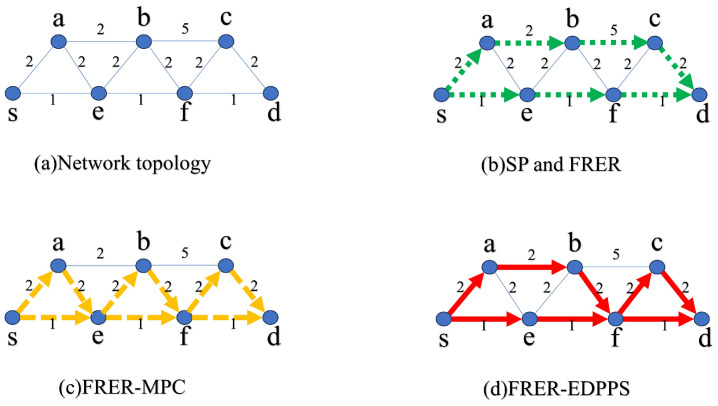
A working example of FRER-EDPPS.

**Figure 4 sensors-24-01738-f004:**
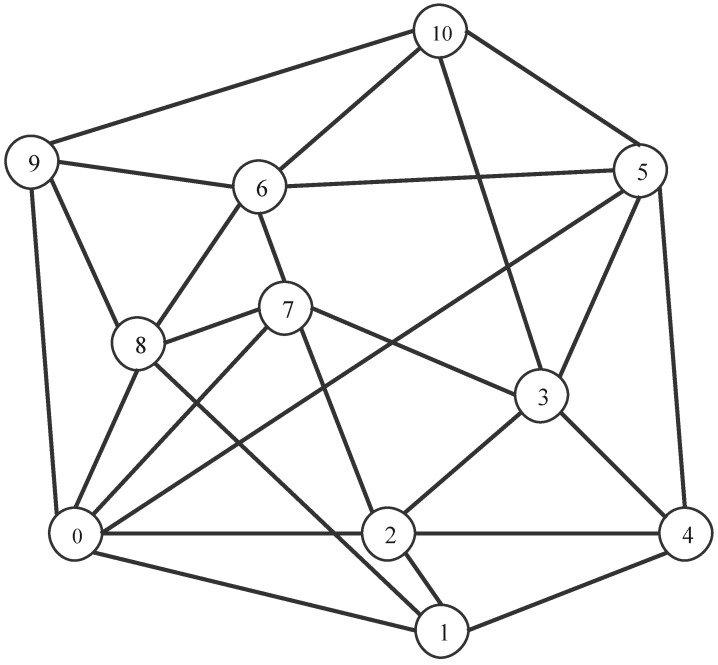
COST-239 network topology.

**Figure 5 sensors-24-01738-f005:**
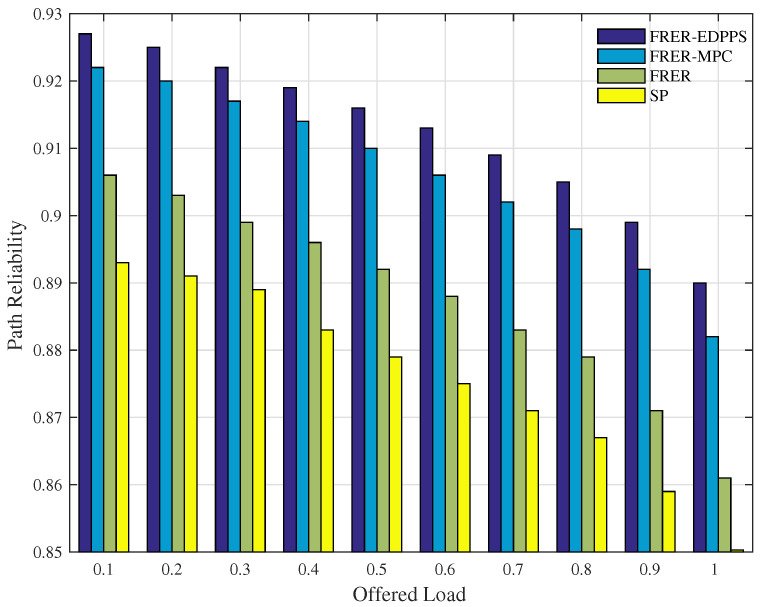
Performance of path reliability under different offered loads.

**Figure 6 sensors-24-01738-f006:**
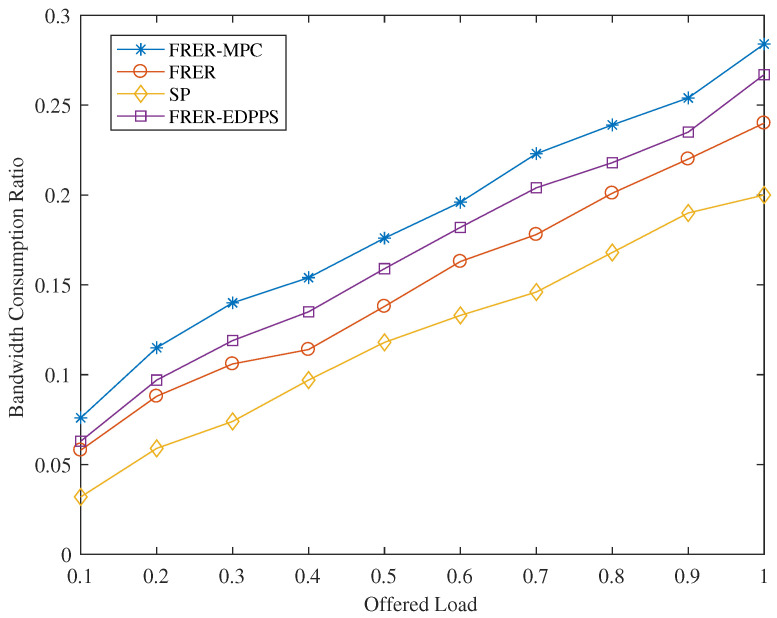
Performance of bandwidth consumption rate under different offered loads.

**Figure 7 sensors-24-01738-f007:**
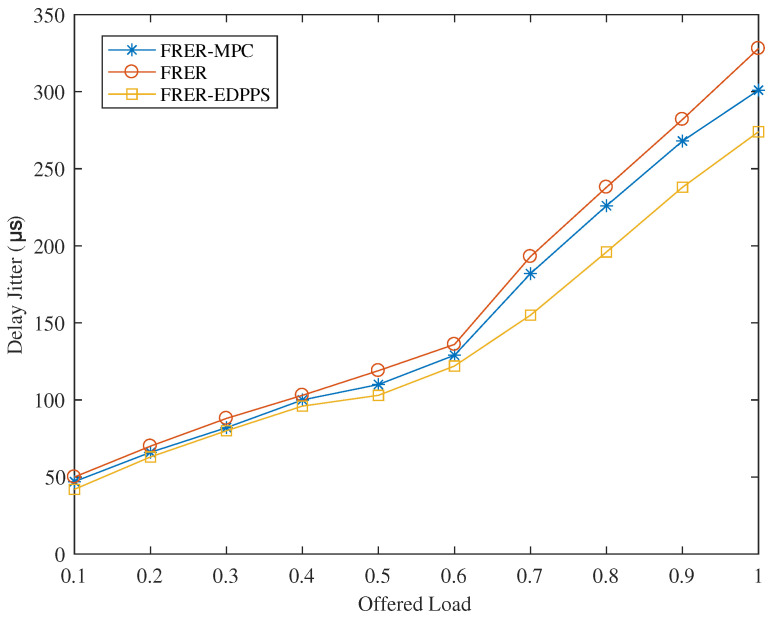
Performance of delay jitter under different offered loads.

**Figure 8 sensors-24-01738-f008:**
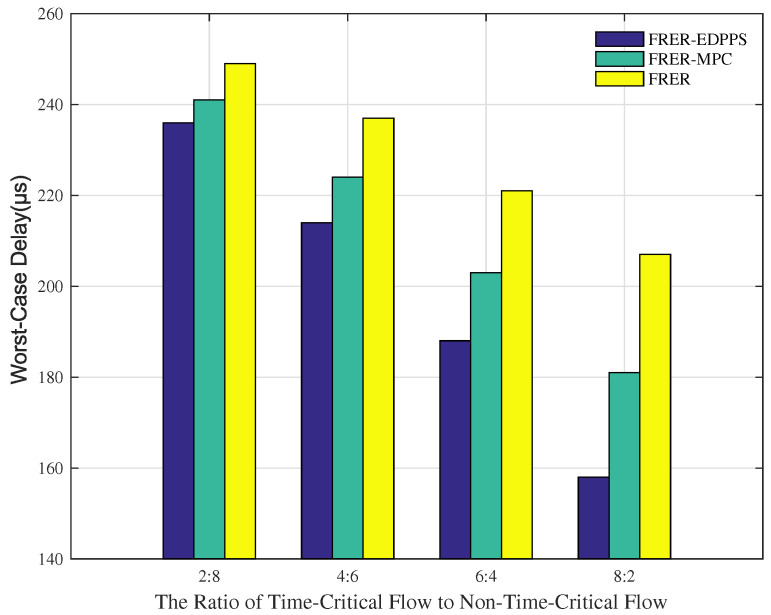
Performance of worst-case delay under different ratios of time-critical flow to non-time-critical flow.

**Figure 9 sensors-24-01738-f009:**
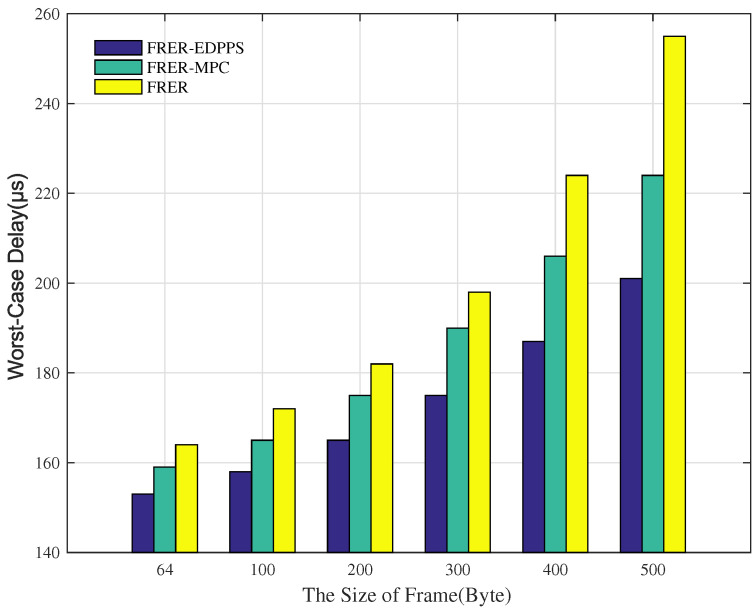
Performance of worst-case delay under different frame sizes.

**Table 1 sensors-24-01738-t001:** Simulation parameters.

Parameters	Distribution	Value
Bandwidth (Gbit/s)	-	1
Link failure	Uniform	[0, 0.1]
Node failure probability	Uniform	[0, 0.1]
Offered load	-	[0.1, 1]
Number of frames	-	$5\times 10^{4}$
Frame size (Bytes)	Uniform	64–500
Frame arrival process	Poisson	500
Frame interval time (μs))	Negative exponent	10

## Data Availability

The data are available upon request.

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
