# Peer review of "Enhanced FRER Mechanism in Time-Sensitive Networking for Reliable Edge Computing"

_sensors, 2024, doi:10.3390/s24061738_

Round 1

Reviewer 1 Report

Comments and Suggestions for Authors

- abstract needs to define exactly what is the main problem statement that wants to solve here!

- the research methodology needs to describe the detailed of TSN and the strenght and weakness points of it .

- The research did not clarify the importance of calculating link reliability, especially with optical networks because it is known that these networks do not need this degree of reliability, and this is possible for wireless networks.

- also this article needs to focus dijikstra algorithm with multi-objective metrics, the author listed three of them but we dont find use of them in the simulation.

Comments on the Quality of English Language

Moderate editing of English language required

Reviewer 2 Report

Comments and Suggestions for Authors

Dear Authors,

Article is well structured and the topic is interesting. However, following comments should be addressed prior to further processing of the article.  

1)      Refer to title: Authors need to revise title of the article.

2)      Refer to abstract: Check space in “Time-Sensitive Networking(TSN)”. Recheck whole article for such typos.

3)      Refer to abstract: Recheck and updated the last sentence of the abstract,  

4)      Refer to whole article: Is TSN equally beneficial for other computing paradigms like cloud computing, fog computing etc.?   

5)      Refer to section 1: How do authors believe that “However, the storage and forwarding method, as well as the flow transmission mechanism of traditional Ethernet technology, fail to meet the Quality of Service (QoS) demanded by these time-critical real-time applications.”?

6)      Refer to line # 38: What is EC? Is it edge computing? Ensure that all short form are described at their first occurrence in the text.

7)      Refer to figure 1: Ensure that data flow path directions are shown in the figure.

8)      Refer to algorithm 1: Algorithm is not properly formatted.

9)      Refer to figures 7, 8 and 9: SP is not shown in the results shown in figures 7, 8 and 9. Why?  

Good luck.    

Comments on the Quality of English Language

Dear Authors,

Article is well structured and the topic is interesting. However, following comments should be addressed prior to further processing of the article.  

1)      Refer to title: Authors need to revise title of the article.

2)      Refer to abstract: Check space in “Time-Sensitive Networking(TSN)”. Recheck whole article for such typos.

3)      Refer to abstract: Recheck and updated the last sentence of the abstract,  

4)      Refer to whole article: Is TSN equally beneficial for other computing paradigms like cloud computing, fog computing etc.?   

5)      Refer to section 1: How do authors believe that “However, the storage and forwarding method, as well as the flow transmission mechanism of traditional Ethernet technology, fail to meet the Quality of Service (QoS) demanded by these time-critical real-time applications.”?

6)      Refer to line # 38: What is EC? Is it edge computing? Ensure that all short form are described at their first occurrence in the text.

7)      Refer to figure 1: Ensure that data flow path directions are shown in the figure.

8)      Refer to algorithm 1: Algorithm is not properly formatted.

9)      Refer to figures 7, 8 and 9: SP is not shown in the results shown in figures 7, 8 and 9. Why?  

Good luck.    

Reviewer 3 Report

Comments and Suggestions for Authors

1. Summary of contribution

This paper introduces a novel approach to enhance the FRER mechanism in TSN for edge computing, addressing its shortcomings by proposing a method for selecting edge-disjoint path pairs that considers path reliability. It innovates with a path reliability model that assesses both links and nodes, and an EDPPS algorithm, enabling the selection of the most reliable path pairs across any network topology. This methodology ensures high reliability and redundant transmission of time-sensitive frames, addressing the FRER mechanism's previous limitations in considering path reliability and finding disjoint path pairs.

2. Weaknesses and suggestions

Please feel free to correct me if I'm misunderstanding:

(a) In your discussion of prior works [42,43,44] between lines 143 to 153, which relate closely to TSN optimization and reliability. And I also found that your proposed path reliability approach draws significant inspiration from them, particularly [42]. This raises questions about the unique benefits your method offers over these established works. Could you benchmark your algorithm against these references (RAMR [42], PFT-TSN [43], and SMT-based method [44]) to demonstrate its superior performance? Without such a comparison, the practical effectiveness of your proposed solution might remain uncertain.

(b) Your complexity analysis mentioned from lines 330 to 333 suggests using Dijkstra’s algorithm with a complexity of $O(M+Nlog(N))$. However, given that Algorithm 1 requires calculating a shortest path set (as indicated in lines 346 and 349), wouldn't the total computational effort be $O(N(M+Nlog(N)))$? This adjustment implies that the overall time complexity is closer to $O(N^2log(N))$, not $O(Nlog(N))$? This distinction becomes particularly significant in networks with a large number of nodes. Could you clarify how your approach intends to efficiently manage the increased computational demands in such extensive real-world networks?

Round 2

Reviewer 1 Report

Comments and Suggestions for Authors

The author needs to compre his work with mentioned related works such as [41-44].

Comments on the Quality of English Language

The english writing is fine 

Reviewer 3 Report

Comments and Suggestions for Authors

The latest revision and response have comprehensively addressed my initial concerns, and I am eager to see further contributions from your team in this field.
